# InstaTune: Instantaneous Neural Architecture Search During Fine-Tuning

**Sharath Nittur Sridhar**[*]**, Souvik Kundu**[*]**, Sairam Sundaresan**[*]**, Maciej Szankin, Anthony Sarah**
Intel Labs, San Diego, USA

## Abstract

One-Shot Neural Architecture Search (NAS) algorithms often rely on training a hardware agnostic super-network for a domain specific task. Optimal sub-networks are then extracted from the trained super-network for different hardware platforms. However, training super-networks from scratch can be extremely time consuming and compute intensive especially for large models that rely on a two-stage training process of pre-training and fine-tuning. State of the art pre-trained models are available for a wide range of tasks, but their large sizes significantly limits their applicability on various hardware platforms. We propose InstaTune, a method that leverages off-the-shelf pre-trained weights for large models and generates a super-network during the fine-tuning stage. InstaTune has multiple benefits. Firstly, since the process happens during fine-tuning, it minimizes the overall time and compute resources required for NAS. Secondly, the sub-networks extracted are optimized for the target task, unlike prior work that optimizes on the pre-training objective. Finally, InstaTune is easy to "plug and play" in existing frameworks. By using multi-objective evolutionary search algorithms along with lightly trained predictors, we find Pareto-optimal sub-networks that outperform their respective baselines across different performance objectives such as accuracy and MACs. Specifically, we demonstrate that our approach performs well across both unimodal (ViT and BERT) and multi-modal (BEiT-3) transformer based architectures. Additionally, we show that using our approach to jointly optimize for the network architecture and mixed precision quantization policy, yields sub-networks with significantly lower model-size.

## 1 Introduction

Neural architecture search (NAS) [1, 2], has become a popular method to generate optimal deep neural network (DNN) architectures for various computer vision and NLP applications. While NAS helps to automate the process, it trades manual effort for computational cost often making its use prohibitive depending on the size of the dataset(s) and generated architectures [3]. Recent advances in NAS have attempted to decrease associated complexity and computational costs to extend its applicability [4, 5].

One-shot NAS algorithms [6, 7] make use of a *super-network* which is first trained and then used to find Pareto-optimal sub-networks on specified performance metrics such as accuracy and latency. This method requires *only* the super-network to be trained and thanks to the weight sharing principle between it and any sampled sub-network (sharing a fraction of trained weights of the supernet), it does not explicitly need any separate training of a sampled sub-network. The search can then be performed using methods like genetic algorithms [6, 8]. Therefore, the time needed to generate optimal models is significantly reduced. These savings multiply when different networks are needed for different performance metrics or hardware platforms.

---

[*]Authors have equal contribution.

37th Conference on Neural Information Processing Systems (NeurIPS 2023).

However, training an elastic super-network using one-shot NAS can be computationally prohibitive, precluding the generation of optimal models. This gets worse as the size of the super-network and/or training datasets grow. In particular, multimodal models [9, 10] make matters challenging since these networks tend to be larger and require significantly more pre-training data compared to unimodal models. On the other end of the spectrum, we have a wide array of pre-trained models, trained via leveraging knowledge of a large corpus of data. Thus, efficient deployment of NAS in the context of "pre-training and down-stream fine-tuning" is largely an open problem.

To leverage the benefits of NAS and those of pre-trained models' knowledge without incurring huge costs, we present InstaTune. In particular, it bypasses the prohibitive training time that NAS requires and leverages the strong initialization provided by a pre-trained model. Post elastic fine-tuning, it enables search for optimal sub-networks from these models, suitable to meet any target hardware constraint. InstaTune converts any off-the-shelf pre-trained model into a super-network during fine-tuning without adding any additional layer(s) (unlike [6]). Rather, existing layers are made elastic at various model dimensions. InstaTune is both cheap and efficient since fine-tuning is orders of magnitude lighter than pre-training both in training time and compute costs. Further, it can produce a family of optimized networks from off-the-shelf models and existing search techniques. This makes InstaTune a "**plug-and-play**" NAS method, allowing practitioners to choose the framework that works best for them.

We show that InstaTune works remarkably well with both unimodal and multimodal models, mitigating the burden of pre-training super-networks while delivering high performance networks for various hardware requirements. Concretely, our evaluations with BEiT-3 Base, ViT-B/16 and BERT Base show that InstaTune can generate a Pareto-frontier of sub-networks that yield models with similar accuracy as their baseline but requiring up to $53.62\%$ fewer multiply-accumulate operations (MACs). Additionally, we demonstrate that jointly searching for both the DNN architecture and a mixed precision quantization policy on the BERT Base InstaTuned network yields significantly lower model-size when compared to the full-precision and INT8 baselines.

## 2   Related Work

Many recent works use NAS for model optimization and efficient neural architecture design [3, 11, 12, 6, 7]. Zero-shot NAS methods [4, 5, 13] use a low cost proxy score to identify and rank top performing candidate architectures for a given MACs budget, without training their parameters. However, use of such proxies in the context of pre-trained models is not straight-forward. Moreover, after the proxy analysis, these methods often demand training of the 'subnet of choice' from scratch. One-shot NAS approaches [6, 14, 15, 16, 17] focus on training task specific super-networks with a weight-sharing mechanism, which allows for efficient extraction of sub-networks without the computational burden of training individual networks from scratch. While most of the super-network based NAS approaches look at convolutional architectures for image-classification, others [7, 18] have demonstrated it on NLP tasks. Approaches such as NASViT [19], ViTAS [20] and NAS-BERT [18] extend one-shot approaches for transformer based architectures on ViT and BERT. However, these methods require extensive pre-training of the super-network using techniques like progressive-shrinking, making it computational expensive and time-consuming. In contrast to these methods, our approach leverages off-the-shelf pre-trained models to create elastic super-network during the downstream fine-tuning stage, making NAS efficient yet well performing.

## 3   Methodology

In this work, we explore both unimodal and multimodal transformer models, namely BEiT-3, ViT, and BERT. Consider a transformer model $\Phi_S$ with $L$ layers, each with $H$ heads. The layers' multi-head self-attention (MHSA) module takes an input tensor $\mathbf{X}$ with sequence length and embedding dimension as $N$ and $D_{in}$, respectively, that then fed through the Query (Q), Key (K), and Value (V) linear transformation layers to generate intermediate tensor $\mathbf{T}_{mhsa} \in \mathbb{R}^{N \times D_{attn}}$. This finally gets projected to the output tensor $\mathbf{O}_{mhsa} \in \mathbb{R}^{N \times D_{in}}$. The succeeding MLP module the intermediate tensor size is $\mathbf{T}_{mlp} \in \mathbb{R}^{N \times D_{ffn}}$ acting as the output and input of the first and second fully connected (FC) layer, respectively and finally produce output $\mathbf{O}_{ffn} \in \mathbb{R}^{N \times D_{in}}$. To train the super-network, we use an elastic search space comprising the number of layers, number of heads, and intermediate MLP dimensions. We denote the maximum values of these elastic parameters using $L$, $H$, and $D_{ffn}$

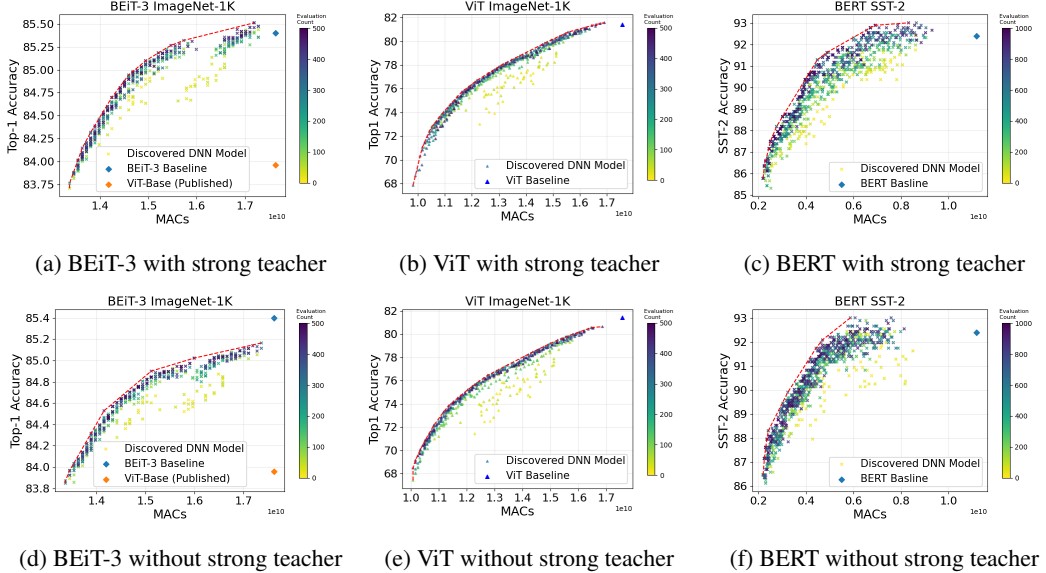

(a) BEiT-3 with strong teacher     (b) ViT with strong teacher     (c) BERT with strong teacher

(d) BEiT-3 without strong teacher    (e) ViT without strong teacher    (f) BERT without strong teacher

Figure 1: Search results for the elastic models generated after InstaTune fine-tuning with and without the strong teacher. For both scenarios we fine-tuned the BEiT-3 Base, ViT-B/16, and BERT Base for 70, 10, and 4 epochs, respectively.

respectively. In particular, we use the baseline pre-trained models as the starting point, and apply the elasticity in the mentioned dimensions to allow sub-network generation during fine-tuning. The fine-tuning loss is,

$$\mathcal{L}_{total} = \alpha \{ \mathcal{L}_{CE}(\Phi_S(\mathbf{X})) + \sum_{i=1}^{M} \mathcal{L}_{CE}(\Phi_S^i(\mathbf{X})) \} + \tag{1}$$
$$(1-\alpha) \{ \gamma * \mathcal{L}_{KL}(\Phi_S(\mathbf{X}), \Phi_T(\mathbf{X}), \rho) +$$
$$\sum_{i=1}^{M} ((1-\gamma) * \mathcal{L}_{KL}(\Phi_S^i(\mathbf{X}), \Phi_S(\mathbf{X}), \rho) +$$
$$\gamma * \mathcal{L}_{KL}(\Phi_S^i(\mathbf{X}), \Phi_T(\mathbf{X}), \rho) \}$$

Here, $\Phi_T$ is a fine-tuned network (on the same downstream task) that has the same architecture configuration as the elastic super-network and $\Phi_S^i$ represents a randomly selected sub-network $i$ to compute the forward pass loss along with the super-network. The coefficients $\alpha$ and $\gamma$ represents the relative strengths of the Cross Entropy (CE) and Kulback-Leibler (KL) Divergence losses, and the presence or absence of a teacher, respectively. $\rho$ is the temperature of the KL-divergence. Note, $\gamma$ is a binary value while $\alpha$ can take any value between 0 and 1. Unless otherwise stated, we keep the weight of CE loss to be 0.3. We always start from a pre-trained unimodal or multi-modal model and only apply the above losses during down-stream fine-tuning. This allows to leverage to leverage the benefits of self-supervised (SSL) pre-training on a large corpus of data and yields multiple sub-network options for inference on down-stream applications.

$M$ in the above Eq. represents the number of sub-networks sampled during each forward pass. We understand there are various sophisticated sub-network sampling methods present in the literature [21]. However, we aim to demonstrate the efficacy of elastic fine-tuning of models, and thus use simple random sampling. We leave the exploration of efficient sampling techniques for future research. The elastic fine-tuning task can be partitioned into three components.

**Low cost fine-tuning.** Often fine-tuning may happen on small devices as well where compute and storage are limited. To focus on the reduced compute budget, we leverage only the the elastic

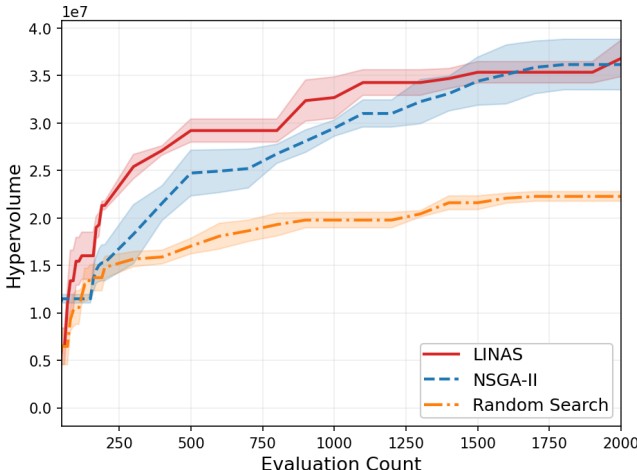

Figure 2: Hypervolume presenting performance of search progression on BERT SST-2 search space using 3 search methods - LINAS, NSGA-II, and Random Search. The higher the hypervolume measures, the better solutions that are being found in terms of both objectives for a given number of evaluated models.

super-network as the teacher to distillation and assume $\gamma = 0$ throughout the fine-tuning. This reduces the forward propagation and storage costs of a teacher of similar size as the super-network. However, to adaptively help the elastic super-network yield higher accuracy, we may allow the $\Phi_T$ to be present during only initial phase of fine-tuning, instead of keeping it active throughout. We term $\Phi_T$ as the 'strong teacher' as it is a fully fine-tuned model on the downstream task.

**High cost fine-tuning.** Here, we assume the fine-tuning is not limited by compute budget and allow the $\Phi_T$ to be present throughout the fine-tuning epochs. This necessitates the use of two forward passes (one for the elastic super-network and one for the strong teacher) to compute the KL div. loss making the fine-tuning costlier. However, later we demonstrate, such presence of $\Phi_T$ allow both the super-network and sampled sub-networks reach better accuracy.

Once the elastic super-network is trained, we use a lightweight iterative NAS (LINAS) [8] to evaluate the multi-objective Pareto frontier. In particular, to reduce the search cost compared to the traditional approaches like NSGA-II, LINAS uses a iterative predictor based approach to come up with better sub-network set during every iteration. Please refer to [8] for further details. Fig. 2 demonstrates the efficacy of LINAS over alternative optimizations including NSGA-II and random search.

# 4 Experimental Evaluation

## 4.1 Experimental Setup

Following our proposed approach outlined in Section 3, we first create and train super-networks for the selected DNN architectures. We then perform a multi-objective sub-network search using the LINAS algorithm proposed in [8], with classification accuracy and MACs as two objectives.

We evaluated our method using ViT-B/16 [22], BERT Base [23] and BEiT-3 Base [10] networks pre-trained on a large data corpus. For ViT and BEiT-3, we chose image classification on ImageNet-1K as the main task. To demonstrate the applicability of our method in other modalities, we also conducted sentiment analysis experiments with BERT on SST-2 [24]. During fine-tuning we use $M = 1$, meaning we use only one randomly sampled sub-network to add to the loss of the super-network. Unless otherwise stated, for ViT and BeiT-3 the elastic dimension values are [11,12], [6,8,10,12], and [2048, 2560, 3072] for $L$, $H$, and $D_{ffn}$, respectively. For BERT these values are [6,7,8,9,10,11,12], [6,8,10,12], and [1024, 2048, 3072], respectively.

## 4.2 Results and Analysis

**Pareto-frontier analysis.** Fig. 1 shows the results of InstaTune for three different models on their respective downstream tasks as accuracy vs. MACs Pareto frontiers. Our method yields multiple

sub-networks that are close to the baseline accuracy while costing significantly fewer MACs. For example, for BEiT-3 trained with a strong teacher, a sub-network with 21.67% fewer MACs can yield an accuracy of 84.32%. This highlights the efficacy of InstaTune as a **plug-and-play** method to yield subnetworks that can be used in resource constrained inference with minimal fine-tuning. The baseline models (the ones without elasticity) did not use any distillation. They are however trained using iso-hyperparameter settings to report their respective accuracies. When InstaTune is used without strong teacher distillation, we see a drop in the accuracy of the sub-networks primarily due to slower convergence. This highlights the need for the teacher in the case when we can not afford to fine-tune a selected subnetwork for additional epochs.

| Model | Sub-networks | Accuracy ↑ | MACs (G) ↓ | $\delta_{MAC}$ ↑ | $\delta_{ACC}$ ↓ |
|---|---|---|---|---|---|
| BEiT-3 Base | Baseline | 85.40% | 17.62 | 0% | 0% |
| | Subnet-1 | 85.32% | 15.73 | 10.72% | 0.09% |
| | Subnet-2 | 84.86% | 14.51 | 17.65% | 0.63% |
| | Subnet-3 | 84.32% | 13.80 | 21.67% | 1.26% |
| ViT-B/16 | Baseline | 81.41% | 17.6 | 0% | 0% |
| | Subnet-1 | 81.59% | 16.90 | 3.97% | -0.22% |
| | Subnet-2 | 81.41% | 16.59 | 5.73% | 0% |
| | Subnet-3 | 80.77% | 15.55 | 11.64% | 0.79% |
| BERT Base | Baseline | 92.40% | 11.17 | 0% | 0% |
| | Subnet-1 | 93.00% | 8.31 | 25.60% | -0.65% |
| | Subnet-2 | 92.43% | 6.41 | 42.61% | -0.03% |
| | Subnet-3 | 91.74% | 5.18 | 53.62% | 0.71% |

Table 1: Performance comparison of different sub-networks (trained with $\Phi_T$ distillation) with the baseline. $\delta_{MAC}$ and $\delta_{ACC}$ is the relative percentage difference in MACs and accuracy, respectively, compared to the baseline.

**Study on the impact of fine-tune epochs.** Fig. 3 shows supernets generated after InstaTune training for different number of epochs. It is noteworthy that despite the consistent improvement with the strong teacher ($\Phi_T$), its impact in improving the accuracy reduces as we fine-tune for longer duration. This directly correlates with $\Phi_T$'s involvement in expediting convergence. However, training for longer allows the model to settle to its learning capacity limit.

**Study on the impact of strong teacher.** Table 2 shows an ablation study with the model ($\Phi_S$) InstaTuned with and without the strong teacher ($\Phi_T$).

| $\Phi_S$ Acc. % w/o $\Phi_T$ | | $\Phi_S$ Acc. % w $\Phi_T$ | | Baseline Acc % after | |
|---|---|---|---|---|---|
| Epoch: 10 | Epoch: 20 | Epoch: 10 | Epoch: 20 | Epoch: 10 | Epoch: 20 |
| 82.22 | 83.84 | 83.15 | 83.95 | 82.62 | 84.70 |

Table 2: BEiT-3 performance InstaTuned for a total of 20 epochs on ImageNet-1k. We report the accuracies after $10^{th}$ and $20^{th}$ epochs for models trained in three different ways. For the model with $\Phi_T$, we only keep the strong teacher for only the first 10 epochs, i.e. keep $\gamma = 1.0$ till $10^{th}$ epoch and make 0.0 after that (Eq. 1).

As seen before, elastic fine-tuning of the super-network converges slower when compared to the baseline fine-tuning. This can be attributed to the variation in loss gradient update directions between the super-network and a random sub-network during each iteration. To resolve this, we can either fine-tune the super-network for more epochs, or we can use $\Phi_T$. As seen in the Table 2, the model with $\Phi_T$ converges faster compared to the baseline and the one without $\Phi_T$.

**Impact of having different elastic search spaces.** A larger search space can provide more sub-network options during search. Fig. 4 shows the Pareto front with two different search spaces (one larger compared to the other). The one with the larger search space yields more models that have lower MACs. If we focus on the sub-networks with higher MACs, their performance remains similar to ones generated over a smaller search space. This hints at the efficacy of InstaTune for larger search spaces.

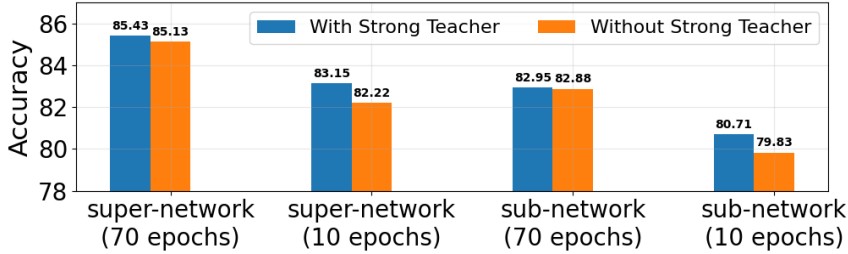

Figure 3: Accuracy comparison for super-networks and a selected sub-network trained at different fine-tune epochs, with and without the strong teacher for BEiT-3. The selected sub-network configuration has $L$, $H$, and $D_{ffn}$ (for all layers) values of 10, 10, and 3072, respectively.

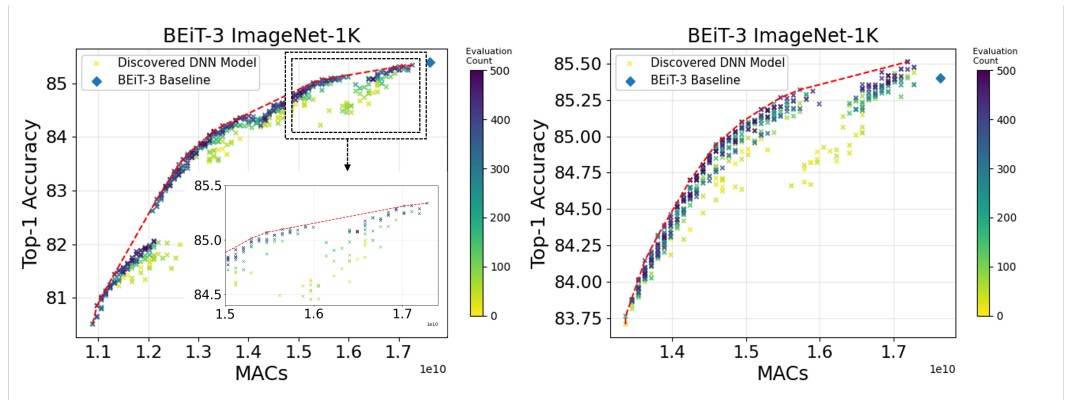

Figure 4: Pareto front comparison for two search spaces having different layer counts for BEiT-3. a) $L$=[9,10,11,12] and b) $L$=[11,12]. Increasing the search space results in more sub-networks in the lower MACs regime, while also maintaining a minimal drop in accuracy at the higher MACs regime.

**Joint Architecture and Quantization Policy Search.** We perform a joint architecture and quantization policy search to simultaneously optimize for the DNN architecture and a corresponding mixed quantization policy. The LINAS algorithm now explores a combined search space that includes both the network elastic parameters (defined in 4.1) and quantization bit widths for weights and activations of every linear layer. We then perform post-training static quantization for sub-networks with the selected mixed precision quantization policy - namely INT8 and FP32 - using the Intel Neural Compressor library. Figure 5 demonstrates joint search results using the BERT Base super-network InstaTuned for 4 epochs with the strong teacher. We use model-size and accuracy as the two performance objectives for search. The results in Figure 5 indicate that our approach finds sub-networks that have significantly lower model-size (approximately 6x) compared to the FP32 baseline with a similar accuracy of $92.4\%$. Additionally, our approach also yields sub-networks that have a lower model size and higher accuracy when compared to BERT Base with all layers quantized to INT8.

## 5 Conclusions

In this paper, we present "InstaTune", a plug & play elastic fine-tuning system that leverages pre-trained models. Our approach eliminates the pre-training cost of NAS while retaining all its benefits yielding remarkable performance on various downstream tasks. Through extensive experiments on both multi-modal and uni-modal models, we demonstrate the efficacy of InstaTune in yielding sub-networks with reduced MACs while maintaining near-baseline accuracy. Additionally, we demonstrate that joint DNN architecture and quantization policy search using our framework yields sub-networks with significantly lower model-size while maintaining near-baseline accuracy. We plan to extend InstaTune to various other downstream tasks like visual question answering and to other large models. Further, we wish to explore automated search-space selection for models to make our approach truly one-shot.

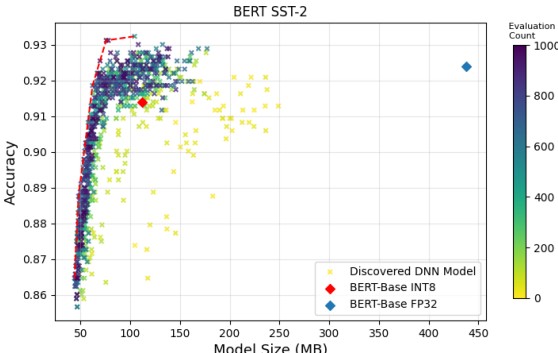

Figure 5: Joint DNN architecture and quantization policy search results for BERT Base InstaTuned with the strong teacher for 4 epochs. The search was performed using LINAS, with accuracy and model-size as the two objectives. Our approach finds sub-networks having significantly lower model size compared to the FP32 and fully quantized INT8 baselines.

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
