# OpenReview forum: "InstaTune: Instantaneous Neural Architecture Search During Fine-Tuning"
_NeurIPS.cc/2023/Workshop/WANT — WANT@NeurIPS 2023 Poster_

### Official Review · Reviewer_1bur · 2023-10-23
**The paper introduces InstaTune, a method for using one-shot NAS for fine-tuning large pre-trained models. The paper proposes a simple distillation based recipe based on the original pre-trained model and shows obtaining subets with high accuracies on downstream tasks.**

**Confidence:** 4

**Review:**

**Originality:**
The authors suggest the use of one-shot NAS for fine-tuning large pre-trained models. The work pushes a new direction in fine-tuning and opens up the application of NAS beyond pre-training. The authors propose a framework based on the elasticity of the existing pre-trained models. This is novel because many existing fine-tuning methods tune subnetworks for downstream tasks. Drawing inspiration for this and using the current model's properties - they propose a distillation based framework, that relies both on the teacher and other randomly drawn subnetworks. They limit the search space to 1 other random network for efficiency and show how they can increase the space (Figure 4) for more efficiency in training.

The authors validate all the different choices in their formulation through comprehensive experiments and report results explaining different design choices.

**Clarity:**
The paper is straightforward to understand. The authors have explained and referenced all the relevant concepts needed to understand the method suggested in this paper.

**Quality**:
The submission is technically sound. An extensive empirical validation very well supports the claims in the papers.

**Significance:**
The paper pushes for a new method by applying NAS for fine-tuning large pre-trained models. This is a relatively non-explored space, and the work has the potential to propel the application of NAS beyond just resource constrained pre-training.

**Weaknesses:**
The authors do not have relevant literature to compare against, given they propose a potential new avenue for fine-tuning. It will be interesting to see how this holds regarding MACs for fine-tuning vs efficient fine-tuning methods like LoRA [1] or Adapters [2] on the same downstream tasks. The authors do not have to report these numbers during the rebuttal - just a suggestion for improving the work in the future to solidify the method's potential. The authors do change the value of $\gamma$ empirically during training, making it tricky to scale this to models quickly, as it requires manual tuning to figure out the best schedule for $\gamma$.

**Broader Impact:**
The authors have yet to discuss broader societal or potential negative impacts in the paper, but nothing unethical stands out in or as a result of their work.

**Overall Score:**
7; A good submission; accept.

---

### Official Review · Reviewer_FxH4 · 2023-10-24
**This paper suggests a novel method for super-network generation during Transformer-based models fine-tuning. Achieved results are compared with baseline only, practical value is not clear. My recommendation: REJECT.**

**Confidence:** 5

**Review:**

## Summary

This paper suggests a novel method for super-network generation during Transformer-based model fine-tuning. Specifically, the authors build their method during the model fine-tuning, by that achieving large benefits compared to full pre-training. They demonstrate a way in which super-network can be trained during fine-tuning and demonstrate experimental results. The authors state that they achieve Pareto-optimality with the proposed method. They make conclusion about simplicity of integrating their method into the existing training pipelines.

## Strengths

1. Writing is easy to follow, language is good
2. Support of Transformer architecture for both uni- and multi-modal models


## Weaknesses

1. **Poor selection of baseline, important information is missed**
   - no limitations section
   - no risks section

   Without this information, it is hard to evaluate the impact of the work on the current state-of-the-art. As for baseline, only full fine-tuning is selected. It is not clear what is exactly proposed. If it is faster or more economical fine-tuning, why it is not compared with LoRA, P-Tuning, (IA)3, etc. If it is a considerable improvement over existing NAS approaches, why it is not selected against their performance. It is clear that there is accuracy drop, however it is not analyzed, limitations are not clear as well. Although ablation study is given, why no analysis is devoted to the number of sampled networks during fine-tuning, no evaluation of super-network properties for the down-stream task fine-tuning.

2. **No way to reproduce results**
   - no link to code to reproduce reported results
   - no information on hardware setup, base training pipelines

   Thus, it allows to question reported results that might bring a wrong impression to future readers. I would highly recommend addressing this item in the future. However, paper is stated to be devoted to practitioners.

3. **Practical value of the work is not clear**
   The only evidence stated in work is logging norms during training that *can be* used for monitoring training. No exact recommendation on how to use it as a metric. Nor there are examples of more clear benefits from the proposed approximations.

4. **Visual elements are not clear**
   Legend is too small, what is a red line on charts (Pareto frontier?), colors are hardly distinguishable.

---

### Meta-Review · Area_Chair_KUUg · 2023-10-27

**Recommendation:** Accept (Poster)
**Confidence:** 3

**Metareview:**

**Strengths:**
* The reviewers agree that this is a well-written paper with core concepts clearly explained.
* Strong evaluation covering both unimodal and multi-modal Transformer architectures.
* Novel contribution: applying NAS beyond resource-constrained pre-training.

**Weaknesses:**
* Missing limitations and risks sections.
* No comparison to PEFT baselines such as LoRA, IA3, etc.
* Missing ablation studies (eg: number of sampled networks).
* Missing information on hardware/training setup and baselines.

This submission has received two very conflicting reviews, and has made the AC’s decision-making very difficult :). Based on my own (short) evaluation, I’m going to give this a borderline accept (poster). However, I recommend assigning additional reviewers to this submission to get a conclusive verdict.

---

### Decision · Program_Chairs · 2023-10-28

**Decision:**

Accept (Poster)

**Comment:**

We thank the authors for their time and contribution to WANT and we are pleased to share that after the reviewing process the paper has been accepted. Congratulations! We encourage the authors to consider reviewers' feedback for the improvement of the camera-ready version. We hope to see you in person at the workshop and brainstorm on efficient training research together!